# Subversion of the Heme Oxygenase-1 Antiviral Activity by Zika Virus

**DOI:** 10.3390/v11010002

**Published:** 2018-12-20

**Authors:** Chaker El Kalamouni, Etienne Frumence, Sandra Bos, Jonathan Turpin, Brice Nativel, Wissal Harrabi, David A. Wilkinson, Olivier Meilhac, Gilles Gadea, Philippe Desprès, Pascale Krejbich-Trotot, Wildriss Viranaïcken

**Affiliations:** 1Université de La Réunion, INSERM UMR 1187, CNRS 9192, IRD 249 UMR PIMIT, Processus Infectieux en Milieu Insulaire Tropical, Plateforme CYROI, 2, rue Maxime Rivière, F-97490 Sainte-Clotilde, France; chaker.el-kalamouni@univ-reunion.fr (C.E.K.); etienne.frumence@univ-reunion.fr (E.F.); sandrabos.lab@gmail.com (S.B.); jonas97480@live.fr (J.T.); wissalharrabi500@yahoo.fr (W.H.); dwilkin799@gmail.com (D.A.W.); gilles.gadea@inserm.fr (G.G.); philippe.despres@univ-reunion.fr (P.D.); 2Université de la Réunion, Inserm, UMR 1188 Diabète Athérothrombose Thérapies Réunion Océan Indien (DéTROI), F-97490 Sainte-Clotilde, France; brice.nativel@gmail.com (B.N.); olivier.meilhac@inserm.fr (O.M.); 3CHU de La Réunion, Saint-Denis de La Réunion, F-97400 Bellepierre, France

**Keywords:** antiviral, heme-oxygenase 1, Zika virus, viral replication

## Abstract

Heme oxygenase-1 (HO-1), a rate-limiting enzyme involved in the degradation of heme, is induced in response to a wide range of stress conditions. HO-1 exerts antiviral activity against a broad range of viruses, including the Hepatitis C virus, the human immunodeficiency virus, and the dengue virus by inhibiting viral growth. It has been reported that HO-1 displays antiviral activity against the Zika virus (ZIKV) but the mechanisms of viral inhibition remain largely unknown. Using a ZIKV RNA replicon with the Green Fluorescent Protein (GFP) as a reporter protein, we were able to show that HO-1 expression resulted in the inhibition of viral RNA replication. Conversely, we observed a decrease in HO-1 expression in cells replicating the ZIKV RNA replicon. The study of human cells infected with ZIKV showed that the HO-1 expression level was significantly lower once viral replication was established, thereby limiting the antiviral effect of HO-1. Our work highlights the capacity of ZIKV to thwart the anti-replicative activity of HO-1 in human cells. Therefore, the modulation of HO-1 as a novel therapeutic strategy against ZIKV infection may display limited effect.

## 1. Introduction

Zika virus (ZIKV) is an emerging mosquito-borne flavivirus (family Flaviviridae), which has become a major medical problem worldwide. In 2007, the first documented outbreak of ZIKV occurred on Yap Island, affecting more than 70% of the population. Subsequently, ZIKV continued to spread in the South Pacific Islands and has largely emerged in the Americas since 2015 [1]. Phylogenetic analysis of viral sequences has identified two main virus lineages, African and Asian, the latter being the main cause of large, current epidemics with millions of cases of infection [2,3].

Classically, the human disease known as Zika fever is characterized by mild flu-like symptoms with fever, maculopapular rash, headache and sometimes conjunctivitis, arthralgia, and myalgia. Epidemiological studies from the recent epidemics have shown that infection due to ZIKV can also cause serious complications in humans such as microcephaly in newborns or Guillain–Barré Syndrome (GBS) in adults [4,5]. Once ZIKV has entered the human body, most commonly through the bite of a mosquito, it targets many types of cells, such as epithelial cells, in order to replicate and produce a viral progeny. Similar to other flaviviruses, the life cycle of ZIKV leads to the release of its single-strand positive sense genomic RNA in the cell cytoplasm where it is translated into a single polyprotein. The polyprotein is cleaved by host and viral proteases into three structural proteins (C, prM/M and E), and seven nonstructural proteins (NS) (NS1, NS2A, NS2B, NS3, NS4A, NS4B and NS5) [6]. The NS proteins initiate the replication process. The viral cycle continues with the production and maturation of envelope proteins, encapsidation, budding, and the release of the virions by exocytosis.

HO-1 is an ubiquitously expressed, stress-inducible enzyme that catabolizes heme into biliverdin, carbon monoxide, and ferrous iron [7]. These products contribute to the cytoprotective effect of HO-1 by acting as anti-oxidant, anti-inflammatory, and immunomodulatory molecules [7]. In addition, it has been shown that HO-1 can display antiviral activity. The upregulation of HO-1 was shown to limit infection by Hepatitis C and B virus (HCV and HBV), Ebola virus (EBOV), human immunodeficiency virus (HIV), and Dengue virus (DENV) [8].

Even if no therapeutic ZIKV control strategy is available to date, it has been previously proposed that the FDA approved drug Hemin could be used for its anti-viral activity, based on its action on HO-1 [9]. However, the direct effect of HO-1 on ZIKV replication has not yet been demonstrated. Here, using cells expressing a molecular ZIKV replicon, we provide evidence that cobalt protoporphyrin (CoPP), an inducer of HO-1, or direct overexpression of HO-1 inhibits ZIKV replication. Unexpectedly, we observe that ZIKV is able to downregulate HO-1 expression as a result of its own replication. Based on these observations, we conclude that the induction of HO-1 to limit ZIKV infection is likely to be ineffective as a therapeutic strategy. 

## 2. Materials and Methods

### 2.1. Virus and Cell Lines

The clinical isolate PF-25013-18 (PF13) of ZIKV has been previously described [10]. Cell lines used in this study included the A549-Dual™cell line (InvivoGen, a549d-nfis), referred to hereafter as “A549 cells”, and the HEK-Blue™ IFN-α/β cell line (InvivoGen, San Diego, CA, USA) which possesses the HEK-293A backbone and is referred to hereafter as “HEK-293A cells”. Both cell lines were cultured at 37 °C with 5% CO_2_ in MEM Eagle medium, supplemented with 10% heat inactivated fetal bovine serum and 2 mmol·L^−1^
l-Glutamine, 1 mmol·L^−1^ sodium pyruvate, 100 U·mL^−1^ of penicillin, 0.1 mg·mL^−1^ of streptomycin and 0.5 µg·mL^−1^ of fungizone (PAN Biotech, Aidenbach, Germany). Additionally, A549 cellular growth medium was supplemented with 10 µg·mL^−1^ blasticidin and 100 μg·mL^−1^ zeocin (InvivoGen) and HEK-293A cell growth medium was supplemented with 30 µg·mL^−1^ blasticidin and 100 μg·mL^−1^ zeocin (InvivoGen). Cells were harvested and stored as frozen pellets for further protein or mRNA analysis.

### 2.2. Generation of ZIKV Replicon by the ISA Method

The production of a ZIKV RNA replicon with GFP as a reporter protein, named ZIKV replicon in the study, was based on the sequence of ZIKV strain MR766 Uganda 47-NIID (Genbank access # LC002520) using the ISA (Infectious Subgenomic Amplicons) method [11]. The design of the viral genome, without the structural protein region, into four viral genomic fragments Z1-PURO, Z2, Z3, and Z4-IRES-eGFP (Figure 1A) was chosen to mimic those used to construct the molecular clone MR766^MC^ [11]. Compared to the conventional method based on the in vitro transcription from a T7 promoter, our construction was designed to allow the in cellulo transcription of recombinant DNA obtained from the four viral genomic fragments Z, by replacing the T7 promoter with a CytoMegaloVirus (CMV) promoter [12]. The fragment Z1-PURO contains the CMV promoter immediately adjacent to the 5′ non-coding region of MR766 followed by the puromycin resistance (PAC) cassette fused in frame to the first 33 amino acids of the ZIKV C protein. It is ended by the porcine teschovirus-1 2A protease and the nucleotide sequence coding for the last 95 C-terminal amino acids of the E protein. The fragments Z2, Z3, and Z4 have been previously described [11]. The fragment Z4-IRES-eGFP contains a cassette containing the Interne Ribosomal Entry Site IRES2 sequence from Clontech followed by the eGFP sequence between the NS5 gene and the 3′NTR from Z4. HEK-293A cells were electroporated with the four fragments Z1-PURO to Z4-IRES-eGFP, using a gene pulser II according to manufacturer protocol (BioRad, Hercules, CA, USA). Cells harboring the ZIKV replicon were selected with puromycin (1 µg·mL^−1^) for at least five days (Figure 1A). The selected cells were kept in culture throughout the experiment (passage 1 to 5). GFP expression was checked periodically by RT-PCR (Figure 1B), cytometry (Figure 1C), as well as by microscopy (Figure 1D). The replicon maintenance was followed by RT-PCR (Figure 1B) and the immunodetection of double-stranded RNA (Figure 1D). Viral protein expression was verified during the different passages by immunodetection (Appendix A).

### 2.3. RNA Isolation and RT-PCR

Total RNA was extracted from cells seeded in 60 mm Petri dishes with the RNeasy Mini Kit (Qiagen, Venlo, Netherlands). RT-PCR was performed using M-MLV Reverse Transcriptase (Invitrogen, Carlsbad, CA, USA) using 500 ng of total RNA and the GoTaq® enzyme (Promega, Madison, WI, USA) according to manufacturer’s recommended procedures [13]. The primers used for RT-PCR were GFP F: 5′-AAGGCTACGTCCAGGAGCGC-3’, R: 5′-CTTGTGCCCCAGGATGTTGC-3’; NS1: F 5′-AGAGGACCATCTCTGAGATC-3’, R 5′-GGCCTTATCTCCATTCCATACC-3’; NS3: F 5′-ATGCACACTGGCTTGAAGC-3’, R 5′-CAGATGCAACCTGATAGGC-3’; RNA PolII: F 5′-GCACCACGTCCAATG-3’, R 5′-GTGCGGCTGCTTCCA-3’; GAPDH: F 5′-GGGAGCCAAAAGGGTCATCA-3’, R 5′-TGATGGCATGGACTGTGGTC-3’. RT-PCR products were visualized by agarose gel electrophoresis. qPCR was performed using the GoTaq^®^ qPCR Master Mix (Promega). All steps were conducted according to the manufacturer’s instructions. The qPCR data were analyzed using the ΔΔC*t* method and results were normalized to GAPDH, which was used as an internal control.

### 2.4. Western Blot Analysis

Cells seeded in 6-well plates were harvested after two washes with Phosphate buffered saline (PBS) containing phosphatase and protease inhibitors (Thermo Fischer Scientific, Waltham, MA, USA) and lyzed with RIPA lysis Buffer (Sigma-Aldrich, St. Louis, MS, USA). Proteins were separated by 12% SDS-PAGE and transferred onto nitrocellulose membranes [14]. The membranes were first blocked with 5% milk in TBS-Tween for 1 h, then incubated with appropriate dilutions of primary antibody at 1:1000 for 2 h. Anti-rabbit immunoglobulin-horseradish peroxidase and anti-mouse immunoglobulin-horseradish peroxidase conjugates were used as secondary antibodies (dilution 1:2000, Vectors). The membranes were incubated with Amersham *ECL Select* or *prime* Western Blotting Detection Reagent (GE Healthcare, Chicago, IL, USA) and exposed to a film or on an Amersham imager 600 (GE Healthcare) or GeneGnome imager (Syngene, Cambridge, UK). Horseradish peroxidase-conjugated anti-rabbit and anti-mouse antibodies were purchased from Vector Labs. The mouse anti-pan flavivirus envelope E protein mAb 4G2 was produced by RD Biotech. Mouse antibody against HO-1was from Abcam and the mouse antibody against α-tubulin, β-tubulin and M2 FLAG were from Sigma–Aldrich. All Western-blot data are representative of three independent experiments.

### 2.5. ISRE/SEAP Activity Quantification

Interferon-sensitive response element (ISRE) promoter activation was evaluated in supernatant of HEK-293A cells expressing the ZIKV replicon using the substrate Quanti-blue (Invivogen) according to the manufacturer’s recommendations to measured secreted embryonic alkaline phosphatase (SEAP) activity. As a positive control, HEK-293A cells were treated for 24 h with recombinant IFN-β (10,000 UI·mL^−1^, Peprotech).

### 2.6. Flow Cytometry Assay 

To detect GFP-expressing cells, cells were harvest after trypsinization and fixed with 3.7% PFA in PBS at room temperature for 10 min. Fixed cells were subjected to a flow cytometric analysis using FACScan flow cytometer (Becton Dickinson, Franklin Lakes, NJ, USA). The percentage of GFP-positive cells was determined using the FlowJo software package.

### 2.7. Cell Immunofluorescence Staining

HEK-293A cells expressing the ZIKV replicon were grown on glass coverslips, fixed and permeabilized for further incubation in monoclonal mouse anti-double-stranded RNA, clone J2 (English & Scientific Consulting Kft, Szirák, Hungary, J2-1104) (1:200) in 1% PBS-BSA and then with Alexa594-conjugated anti-mouse Ig (1:1000). Nuclei were revealed by DAPI staining (final concentration 100 ng/mL). Coverslips were mounted in Vectashield (Vector Labs; Clinisciences, Nanterre, France), and fluorescence was observed using a Nikon Eclipse E2000-U microscope (Nikon, Tokyo, Japan). Images were obtained using the Nikon Digital sight PS-U1 camera system and the imaging software NIS-Element AR (Nikon).

### 2.8. HO-1 Overexpression 

HEK-293A cells expressing the ZIKV replicon were seeded on 6-well plates and transfected with the expression plasmid pcDNA3.1-Neo or pcDNA3.1-HO-1-Flag-Neo encoding the human HO-1 protein using Lipofectamine 3000 (Invitrogen) according to the manufacturer’s instructions. Transfected cells were cultivated in media with G418 to enforce the expression of HO-1. For preparation of pcDNA3.1-HO-1-Flag-Neo, the human HO-1 open reading frame was synthesized and cloned between the KpnI and XhoI restriction sites of the pcDNA3.1-Neo plasmid (Invitrogen) by GeneCust (Ellange, Luxembourg).

### 2.9. Crystal Violet Assay

To assess cell viability, we used the crystal violet staining assay (adapted from Saotome et al. [15]). Briefly, cells seeded in 6-well plates were washed with PBS and then stained with 0.1% crystal violet in PBS for 15 min. The plates were carefully washed with water. Then, 500 µL of 1% sodium dodecyl sulfate was added to each well to solubilize the stain and absorbance was read at 590 nm. 

### 2.10. Statistical Analysis

All values are expressed as mean ± SD and represent at least three independent experiments. Comparisons between different treatments were analyzed using a one-way ANOVA test. Values of *p* < 0.05 were considered statistically significant for a post-hoc Tukey–Kramer test in order to compare treated versus non-treated. RT-qPCR statistical analysis was carried out by the Student’s *t*-test. All statistical tests were performed using the software Graph-Pad Prism version 5.01 (San Diego, CA, USA). Degrees of significance are indicated in the figure captions as follow: * *p* < 0.05; ** *p* < 0.01; *** *p* < 0.001, ns = not significant.

## 3. Results

### 3.1. Generation of a ZIKV Replicon in HEK-293A Cells by the ISA Method.

For enveloped viruses, most antiviral therapies consist of drugs which target a specific viral protein or cellular cofactor that mediates important steps in the viral life cycle, including viral entry, fusion, replication or egress. Antiviral activity of HO-1 against flaviviruses results in the inhibition of replication [9,16]. To address the effect of HO-1 on ZIKV replication in epithelial cells, independently of other steps in the viral life cycle, we produced HEK-293A cells stably expressing and replicating an RNA molecule acting as a ZIKV replication reporter, i.e., a ZIKV replicon (Figure 1A). 

To validate the maintenance of a ZIKV replicon in the puromycin selected cells, we checked the presence of RNA molecules encoding for NS1, NS3 and the GFP reporter gene by RT-PCR (Figure 1B). These observations were further supported by the expression of NS1 protein (Appendix A). The presence of the GFP reporter gene in the construct allowed us to follow the ZIKV replicon expressing cells by flow cytometry. To ensure that the replicon was maintained in cells, we used Ribavirin, a synthetic nucleoside that inhibits RNA virus replication and blocks their nucleic acid synthesis [17]. The decrease of the GFP signal observed with ribavirin likely corresponds to the effect on self-replication of the replicon (Figure 1C) and has been used elsewhere as direct evidence previously used to validate the function of viral replicons [12,18]. We then confirmed the presence of double-stranded RNA (dsRNA), which corresponds to the active replication of the ZIKV replicon, by immunodetection with the J2 antibody (Figure 1D). Cells expressing the GFP reporter gene were also positive for the J2 staining. Lastly, the presence of dsRNA should trigger an antiviral response through type I interferon production. Since ZIKV replicon cells provide a reporter gene with an ISRE fused to secreted embryonic alkaline phosphatase (SEAP), we were able to confirm the IFN-β response by the measure of SEAP activity in the cell media (Figure 1E). This is additional evidence that the self-replicating system specific to the viral replicon is maintained. The ZIKV replicon was able to assemble and replicate autonomously in the ZIKV replicon cells.

### 3.2. HO-1 Reduces ZIKV RNA Replication

To investigate the effect of HO-1 on ZIKV replication, we first treated HEK-293A cells expressing ZIKV replicon with CoPP, an inducer of HO-1. We verified the upregulation of HO-1 in a dose-dependent manner with increasing concentrations of CoPP (Figure 2A). Under these conditions, no cytotoxic effects were observed using neutral red assay (data not shown). 

We then assessed the replication efficiency of the ZIKV replicon by measuring the percentage of GFP-positive cells by flow cytometry analysis (Figure 2B). We found that CoPP treatment significantly reduced the percentage of GFP-expressing cells at all tested doses and noted that CoPP was able to inhibit ZIKV replicon replication as efficiently as ribavirin.

CoPP is known to upregulate the expression of genes that are under the control of the Antioxidant Response Element (ARE-driven gene) including HO-1 and NQO1 [19]. To investigate a direct involvement of HO-1 in the CoPP-induced inhibition of ZIKV replicon, we quantified the effect of an overexpression of HO-1 in our system. HEK-293A cells with ZIKV replicon were transfected with a plasmid pcDNA3.1-HO-1-Flag-Neo and checked for HO-1 overexpression (Figure 2C). As previously seen with CoPP treatment, the percentage of GFP-positive cells decreased upon HO-1 overexpression (Figure 2D).

Given that the system used to generate the replicon requires the use of the CMV promoter and since it cannot be excluded that the initial DNA fragments may provide constitutive expression of the GFP, we verified that the effect of HO-1 was specific to the replicon and could not be exerted on a CMV dependent transcription. To support an effect of HO-1 induction or overexpression on the replicon and not on such a phenomenon, we confirmed that there was no change in the constitutive expression of GFP under CMV promoter (Appendix A).

To further characterize the effect of HO-1 on the ZIKV replicon, the transfected cells were continuously cultured in media containing G418 and puromycin for 7 days to enforce the expression of HO-1 and ZIKV replicon, respectively. The puromycin N-acetyltransferase (PAC) expression is directly linked to replication of ZIKV replicon. If HO-1 had a negative effect on the upkeep of the ZIKV replicon, we expected a progressive reduction of the puromycin resistance, leading to a growth defect (Figure 2E). Using the crystal violet assay, we observed a massive cell death in the HO-1 overexpressing cell, which is related to a capacity of the enzyme to downregulate the replication of the ZIKV replicon (Figure 2F). Notably, co-expression of a plasmid carrying the PAC gene (pSilencer-puro) and the plasmid pcDNA3.1-HO-1-Flag-Neo does not induce cell death upon selection with puromycin and G418 (Appendix A). 

The results are consistent with available data on the protective effect of HO-1 on ZIKV [9]. In order to be sure that HO-1 was also able to reduce the ZIKV RNA replication during infection in the same assay system as the ZIKV-replicon, i.e., the HEK-293A cells. These backbone cells overexpressing HO-1 were infected with the recombinant ZIKV-GFP [11]. We did notice a reduction in the number of infected cells and a similar reduction was observed in cells treated with CoPP (Appendix A).

Taken together, all of these observations support the inhibition of ZIKV replication by the action of HO-1.

### 3.3. ZIKV Inhibits HO-1 Expression 

HO-1 is known to be rapidly induced for cellular protection under various stresses, including many types of viral infection. Classical swine fever virus (CSFV) or human cytomegalovirus (HMCV) induces an increase in HO-1 expression [20,21]. Conversely, HIV, HCV and Bovine viral diarrhea virus infections are associated with a down-regulation in HO-1 expression [19,22,23]. To determine whether ZIKV interferes with the expression of HO-1, we analyzed whether the level of HO-1 protein was modulated in HEK-293A cells expressing the ZIKV replicon. We found that HO-1 was no longer detectable at the protein level (Figure 3A) and was reduced at the mRNA level (Figure 3B). qPCR values indicate approximately a halving in the number of transcripts (Figure 3C). This suggests that the regulation of HO-1 expression may take place at both transcriptional and post-transcriptional levels upon replication of the ZIKV replicon and/or expression of NS proteins. To confirm that the observed deregulation of HO-1 was indeed a property of ZIKV, A549 cells that are more susceptible to ZIKV than HEK 293A were chosen to be infected with a clinical isolate of ZIKV (ZIKV-PF13) [10] and assessed for HO-1 expression at different times post-infection. HO-1 protein was markedly reduced 24 h post-infection (Figure 3D). It should be noted that molecular clones representing viruses of the African ancestral strain (MR766) or the Brazilian epidemic strain (BR15) were also able to decrease HO-1 expression in another cell line model (Appendix A). In addition, we could confirm a slight significant decrease in HO-1 expression at the level of quantified transcripts in ZIKV-infected A549 cells (Figure 3E,F).

### 3.4. Inhibition of HO-1 Induction During ZIKV Infection 

The inhibiting effect of ZIKV on HO-1 expression may be a limitation in the use of an HO-1 inducer for antiviral purposes [9]. We first tested whether HO-1-induction by CoPP was able to inhibit viral growth of the epidemic ZIKV-PF13 strain. The effect of HO-1 induction by CoPP on viral growth was followed by adding CoPP 2-h post-infection with ZIKV-PF13 at multiplicity of infection (MOI) of 0.1 or 1 (Appendix A). In this experiment, induction of HO-1 with CoPP only decreased the viral progeny production of ZIKV-PF13 at lower MOI. This observation and the down-regulation of HO-1 upon ZIKV infection observed above suggest that the inhibition of HO-1-induction depends on the viral load. To test this hypothesis, we evaluated whether ZIKV was able to counteract the induction of HO-1 by CoPP. We infected A549 cells for 2 h with ZIKV-PF13 at different MOIs and then induced HO-1 expression with CoPP for an additional 16 h. We can see that, despite the presence of CoPP, ZIKV inhibited HO-1 protein level in an MOI-dependent manner (Figure 4A). The ability of ZIKV to inhibit HO-1 induction in response to CoPP at the protein level was not correlated to a significant decreased in HO-1 at the mRNA level (Figure 4B,C). Indeed, ZIKV induces a more robust translational or post-translational downregulation of HO-1 than a transcriptional or post-transcriptional regulation of HO-1 expression.

## 4. Discussion and Conclusions

In the recent years, HO-1 has gained increasing interest as a potential “super cytoprotective agent”. Much work has been done to provide a therapeutic value for HO-1 and to find ways to activate or modulate its expression and activity. Research in this field shows a strong interest in molecules capable of inducing HO-1, including various natural substances such as curcumin [24,25]. Numerous observations have also supported a major role of HO-1 in the control of viral infections, notably through the direct inhibition of viral replication. A role of HO-1 has been described in the case of Dengue after induction by lucidone, a plant extract [26], and the use of HO-1 induction by Hemin has been proposed as a novel modality for developing new therapeutic strategies against ZIKV infection [9]. 

In the present study we provide evidence that HO-1 induction by CoPP or its direct overexpression is able to inhibit ZIKV replication (Figure 2, Appendix A). Our results, therefore, support a beneficial effect of HO-1 in the context of ZIKA pathology which would justify renewed efforts to find effective HO-1 inducers for therapeutic purposes. The modulation of ZIKV growth by HO-1 is related to an effect on replication (Figure 2). It has been previously shown that biliverdin, a product of heme degradation by HO-1, is able to inhibit DENV replication through an interaction with NS2B/NS3 protease activity [16]. This mechanism can be conserved during ZIKV infection and needs to be confirmed by an in vitro assay of ZIKV NS2B/NS3 protease activity.

To our knowledge, this is the first report showing that ZIKV infection modulates HO-1 expression (Figure 3 and Figure 4) in order to limit the antiviral effect of this cytoprotective protein (Figure 5). This result appears to be inconsistent with the effect recently observed with the FDA-approved formulation of Hemin which, by inducing HO-1, inhibits viral growth during ZIKV infection [9]. However, in this report, the formulation of Hemin was added at 100 µM, 24 h before the infection with ZIKV at low MOI (0.01). We therefore assume that under these drastic conditions the virus is not able to counteract the effect of HO-1 by blocking its translation or post-translational regulations. These differences would rather suggest that HO-1 targeting can only work in a preventive but not curative way for the control of ZIKV infection.

While our study confirms that HO-1 induction could be an effective cell-based strategy to control ZIKV infection, it highlights the ability of the virus to target and counteract this antiviral response (Figure 5). Therefore, if HO-1 induction is to be maintained as a promising strategy in the fight against ZIKV infection, it will be necessary to understand at what level ZIKV interferes with HO-1 expression. Nrf2 (the nuclear factor erythroid-related factor 2) is an important transcription factor that activates HO-1 expression through ARE elements located in the HO-1 promoter gene [27]. HO-1 down-regulation during ZIKV replication could occur through the modulation of Nrf2 transcription factor activity or expression. To the best of our knowledge, several viruses (HBV, HCV, DENV, HIV, Respiratory syncytial virus, Marburg virus) induce Nrf2 activation and the subsequent upregulation of the antioxidant responses genes such as NQO1, GSPT2 and HO-1 [28]. The control of HO-1 expression by ZIKV seems quite outstanding and therefore necessitates an understanding of its mechanism of action for the inhibition of the Nrf2/ARE pathway, which results in decreased host defenses against viral infection. This may be extremely important in the case of viruses’ co-circulation (such as DENV during ZIKV outbreak in Brazil in 2015), as ZIKA virus infection may lower cell antiviral defense resulting in exacerbated viremia of the second infecting virus.

In addition, during the induction of HO-1, the effect of ZIKV on HO-1 expression is more pronounced on the level of the protein than of mRNA. This last point therefore suggests that the virus may interfere with the translation of the HO-1 messenger or the degradation of the the HO-1 protein. As it has been previously shown that ZIKV infection can trigger Endoplasmic-Reticulum stress (ER stress) [29] and that HO-1 can be degraded through the Endoplasmic-Reticulum-Associated protein Degradation (ERAD), a proteasome pathway induced during ER stress [30,31], it will be of interest to test if ZIKV downregulation of HO-1 is a combination of the ERAD pathway during ZIKV infection and an interference of ZIKV with Nrf2 transactivation of ARE elements in the HO-1 promoter. While many unanswered questions remain, these observations suggest that a better understanding of ZIKV pathogenicity (such as that associated with the Brazilian epidemic in 2015) may be achieved through a mechanistic understanding of HO-1 inhibition. Here we have established a cellular system to conduct such studies and have observed a crosstalk between HO-1 expression and ZIKV replication (Figure 5), suggesting that HO-1 targeting treatments may be of limited therapeutic efficacy against ZIKV infection.

## Figures and Tables

**Figure 1 viruses-11-00002-f001:**
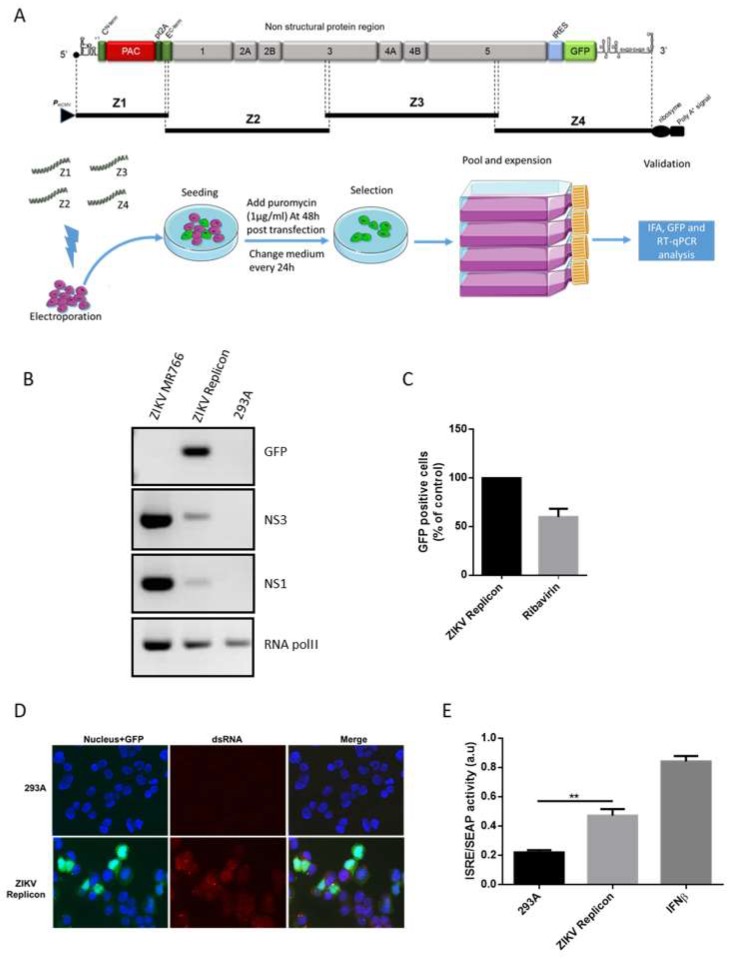
Generation and validation of ZIKA Virus (ZIKV) replicon in HEK-293A cells. (**A**) Schematic representation of overlapping fragments Z1 to Z4 covering of ZIKV replicon. Below, the flow chart representing the design of the experiment. (**B**) GFP, ZIKV NS3, ZIKV NS1 and RNA pol-II mRNA expression assessed by RT-PCR in ZIKV-infected A549 cells, ZIKV replicon cells and HEK 293A cells. (**C**) Cytometry monitoring of GFP after ribavirin treatment. (**D**) Fluorescence microscopy images of ZIKV replicon cells (GFP positive) after immunostaining of dsRNA with J2 antibody (red). (**E**) ISRE/SEAP activity evaluated in HEK 293A cells and ZIKV replicon cells. As positive control, cells were treated for 24 h with recombinant IFN-β (10,000 UI·mL^−1^). ** *p* < 0.01.

**Figure 2 viruses-11-00002-f002:**
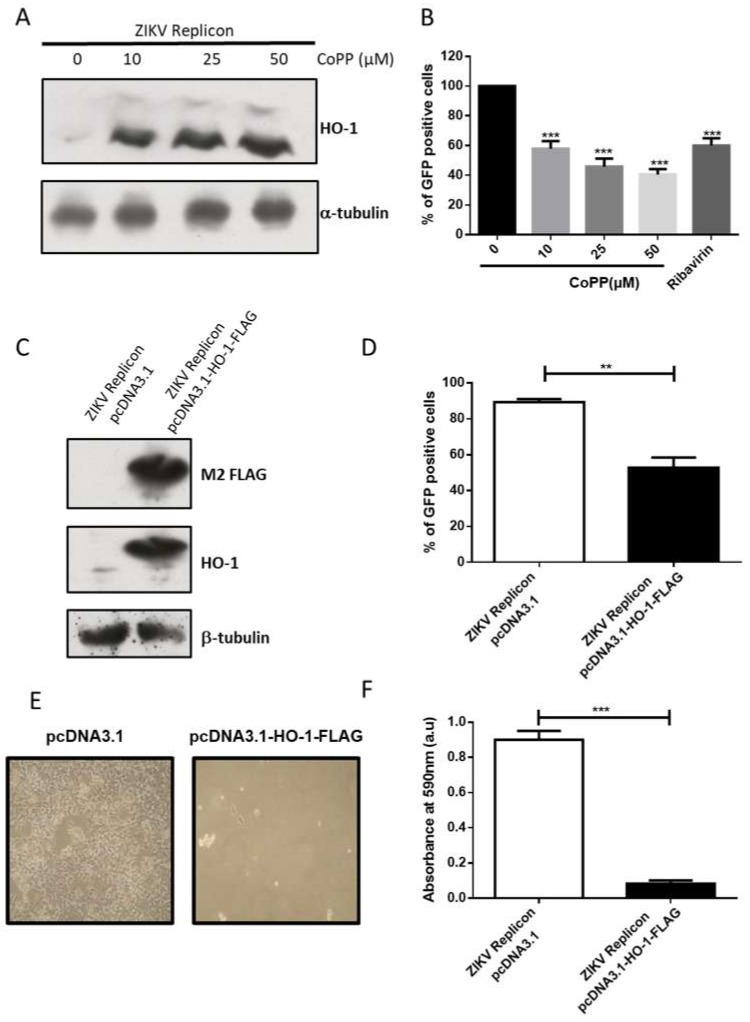
ZIKV replicon expression is inhibited by HO-1 induction or overexpression. (**A**) HO-1 protein expression was assessed by Western blot in HEK-293A cells expressing ZIKV replicon after treatment with several doses of CoPP for 20 h. Antibody against α-tubulin served as the protein loading control. (**B**) The percentage of GFP-expressing cells was analyzed by flow cytometry assay in ZIKV replicon cells after treatment with different concentrations of CoPP for 20 h. As positive control, cells were treated for 24 h with 40 µg·mL^−1^ (164 µM) of ribavirin. *** *p* < 0.001. (**C**) Overexpression of HO-1 in ZIKV replicon cells transfected with pcDNA3.1-HO-1-FLAG-Neo was assessed by Western blot using the anti-FLAG M2 and antibody against HO-1. Antibody against β-tubulin served as protein loading control. (**D**) The percentage of GFP-expressing cells was analyzed by flow cytometry assay in ZIKV replicon cells after transfection with pcDNA3.1-HO-1-FLAG encoding the human HO-1 protein or with pcDNA3.1. ** *p* < 0.01. In (**E**) and (**F**), cell viability was observed by optical microscopy or crystal violet staining respectively in ZIKV replicon cells transfected with pcDNA3.1 or pcDNA3.1-HO-1-FLAG after 7 days of treatment with G418 and puromycin to respectively allow the expression of HO-1 and the expression of ZIKV replicon. *** *p* < 0.001.

**Figure 3 viruses-11-00002-f003:**
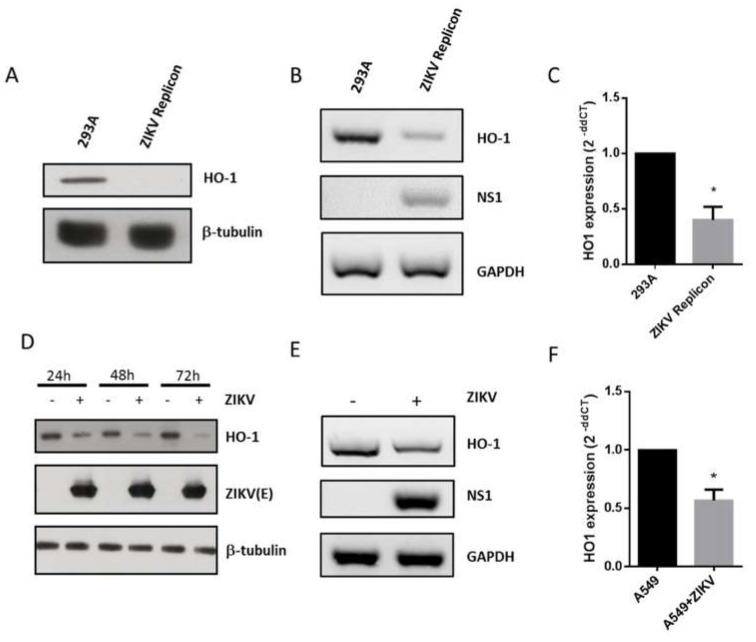
ZIKV replication and ZIKV infection decrease HO-1 protein and mRNA levels. (**A**) Western blot analysis of HO-1 protein expression in HEK 293A cells and ZIKV replicon cells using anti-HO-1. (**B**) RT-PCR analysis of HO-1, ZIKV NS1 and GAPDH mRNA expression in HEK 293A cells and ZIKV replicon cells. (**C**) RT-qPCR analysis of HO-1 expression in HEK 293A cells and ZIKV replicon cells. A549 cells were infected with ZIKV-PF13 at multiplicity of infection (MOI) of 5. In (**D**), HO-1 and ZIKV-E protein expression were analyzed by Western blot using anti-HO-1 and anti-ZIKV-E 4G2 antibody. Antibody against β-tubulin served as protein loading control. In (**E**), HO-1, ZIKV NS1 and GAPDH mRNA expression were analyzed by RT-PCR 24-h post infection. In (**F**) RT-qPCR analysis of HO-1 expression in A549 cells infected with ZIKV-PF13 24-h post-infection. * *p* < 0.05.

**Figure 4 viruses-11-00002-f004:**
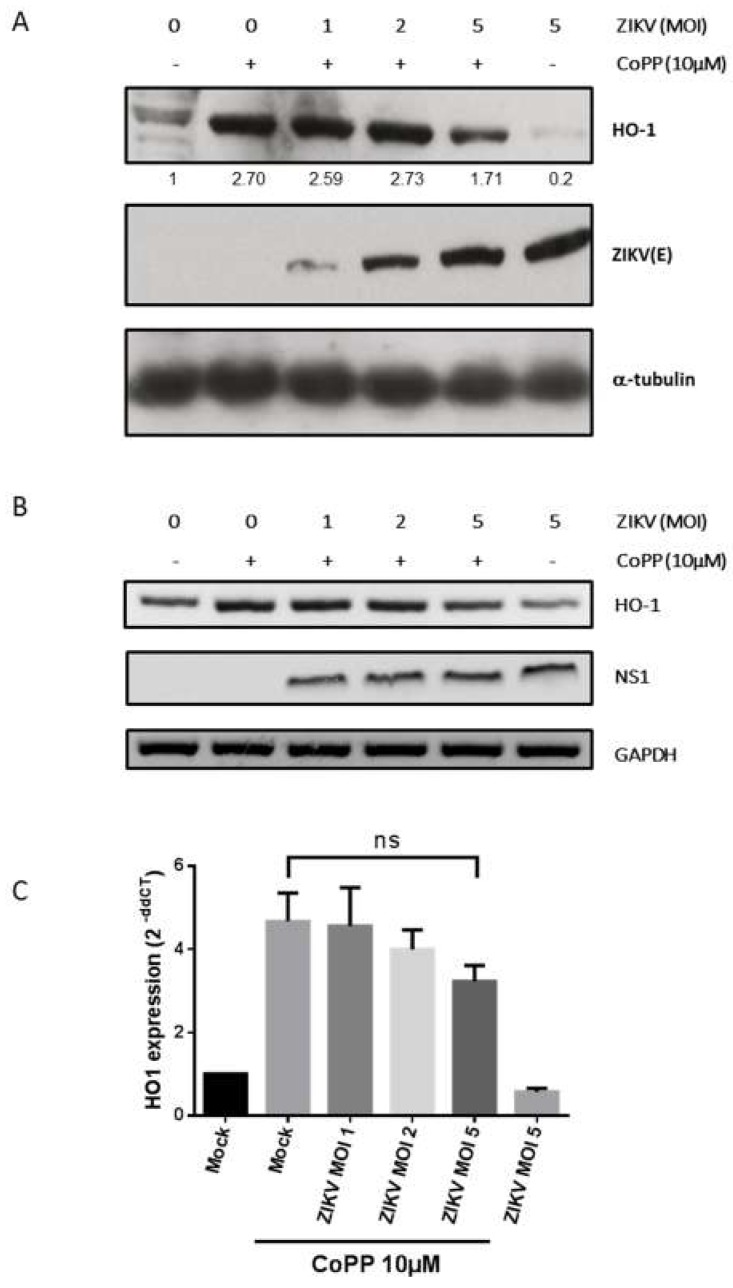
ZIKV infection decreases HO-1 induction mediated by CoPP at the protein and the mRNA levels. A549 cells were infected with ZIKV-PF13 at several multiplicity of infection (MOI) and then treated 2 h post-infection with CoPP for 16 h. In (**A**), HO-1 and ZIKV-E protein expression were analyzed by Western blot using anti-HO-1 and anti-ZIKV-E 4G2 antibody. Antibody against α-tubulin served as protein loading control. Quantification was done with the ImageJ software. In (**B**), HO-1, ZIKV NS1 and GAPDH mRNA expression were analyzed by RT-PCR. In (**C**) RT-qPCR analysis of HO-1 expression. ns = not significant.

**Figure 5 viruses-11-00002-f005:**
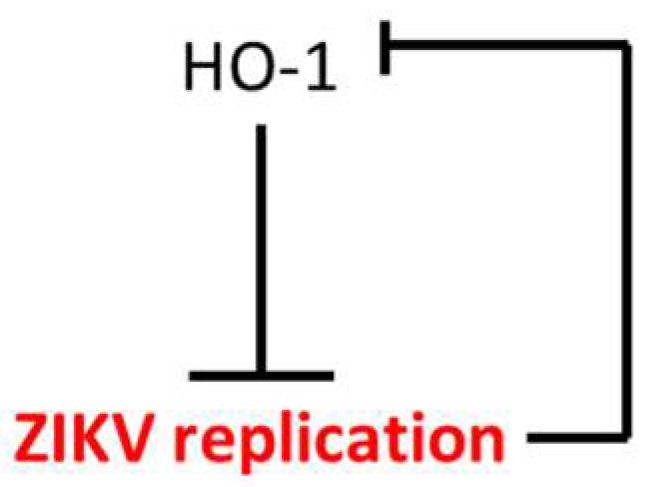
The model of the crosstalk between HO-1 and ZIKV. HO-1 induction inhibits ZIKV at the replication level. ZIKV growth downregulates HO-1 level through its own replication.

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
