# Peer review of "Subversion of the Heme Oxygenase-1 Antiviral Activity by Zika Virus"

_viruses, 2018, doi:10.3390/v11010002_

Round 1

Reviewer 1 Report

Subversion of the Heme Oxygenase-1 antiviral activity by Zika virus (Kalamouni et al)

The manuscript describes the role of Heme oxygenase-1 (HO-1) in modulating ZIKV replication.The study concludes that HO-1 is not efficient in controlling ZIKV. Authors have used a ‘replicon’ system and wild type ZIKV in their study and obtained rather conflicting results. Using cells expressing their modified version of ZIKV replicon the authors found that Cobalt protoporphyrin (CoPP) which is an inducer of HO-1 is able to inhibit ZIKV replication. Similarly, overexpression of HO-1 was also able reduce ZIKV replication in the ‘replicon cells’.

The manuscript reports a clear analysis of the effect of HO-1, however the replicon experiments are not complete and executed poorly. The major concern is the reliability of replicon system the authors used in this study. ZIKV genome specifically very unstable. Thus far all the replicon systems published for flaviviruses are constructed and characterized in vitro and used for further studies. Also in their system the GFP is under the control of an IRES and the puromycin is under the control of CMV promoter. Subgenomic replicons are not to be used under CMV promoter if these are to be used in replication studies. This experiment itself is not done properly and the additional experiments and conclusions derived based on this cell line is unreliable. A replicon should be able to self replicate and produce multiple copies of RNA genome and viral RNA and proteins that can be detected by qRT and IFA respectively. For reliability the authors should show the effect using in vitro transcribed viral RNA from replicon followed by transfection.

To address this concern it is necessary to show the image of GFP expressing cells and western analysis using ZIKV replication proteins such as NS3 or NS5 in figure 2. Figure 1 is showing the RNA expression of different regions of the genome. It is also necessary to show the increase in viral RNA from replicon as a function of time. However, the presence of CMV promoter and constitutive expression of mRNA complicate the use of replicon and interpretation of data reported in this study.

It will be useful if the authors used WT virus in the experiment discussed in figure 2. A more reliable conclusion can be made with respect to the overexpression of HO-1 in cell lines followed by virus infection. Since there is conflicting data from replicon and virus, in order to have reliable results, use virus and replicon in the same assay system and compare results. As is, the authors fail to confirm that HO-1 is down regulating ZIKV replication.

Fig 4:  The effect of ZIKV MOI in controlling the transcription of HO-1 is not clear. There is RNA expression at MOI 1-5. For quantitative analysis, authors will have to use qRT to determine the RNA copy numbers of viral RNA and HO-1 RNA. As is, it does not look like transcriptional control. Similarly, in Fig 3B there is mRNA expression of HO-1 but no protein is seen in 3A. Along with the data shown in S2, the protein expression of HO-1 is negatively affected.

Author Response

Reviewer1

Subversion of the Heme Oxygenase-1 antiviral activity by Zika virus (Kalamouni et al)

The manuscript describes the role of Heme oxygenase-1 (HO-1) in modulating ZIKV replication.The study concludes that HO-1 is not efficient in controlling ZIKV. Authors have used a ‘replicon’ system and wild type ZIKV in their study and obtained rather conflicting results. Using cells expressing their modified version of ZIKV replicon the authors found that Cobalt protoporphyrin (CoPP) which is an inducer of HO-1 is able to inhibit ZIKV replication. Similarly, overexpression of HO-1 was also able reduce ZIKV replication in the ‘replicon cells’.

The manuscript reports a clear analysis of the effect of HO-1, however the replicon experiments are not complete and executed poorly. The major concern is the reliability of replicon system the authors used in this study. ZIKV genome specifically very unstable. Thus far all the replicon systems published for flaviviruses are constructed and characterized in vitro and used for further studies. Also in their system the GFP is under the control of an IRES and the puromycin is under the control of CMV promoter. Subgenomic replicons are not to be used under CMV promoter if these are to be used in replication studies. This experiment itself is not done properly and the additional experiments and conclusions derived based on this cell line is unreliable. A replicon should be able to self-replicate and produce multiple copies of RNA genome and viral RNA and proteins that can be detected by qRT and IFA respectively. For reliability the authors should show the effect using in vitro transcribed viral RNA from replicon followed by transfection.

To address this concern it is necessary to show the image of GFP expressing cells and western analysis using ZIKV replication proteins such as NS3 or NS5 in figure 2.

Figure 1 is showing the RNA expression of different regions of the genome. It is also necessary to show the increase in viral RNA from replicon as a function of time.

We would first like to thank the reviewer for their comments, which helped to consolidate our experimental model and the reliability of our results.

One of the objectives of the study was to develop a system for the study of ZIKV replication. Our experience of the ISA method, efficient for virus production, (Gadea et al., Virology 2016, Bos et al. Virology 2018, Ref 11 and 12, led us to implement it in order to produce cells expressing an autonomous replicon system for ZIKV. Unfortunately, we are unable, within the time limits set, to compare ZIKV replicons obtained by the ISA method with those that would be generated by the classical technique using in vitro transcribed RNA, followed by transfection. Due to the comments, we have added a flow chart for the implementation of our replicon system for more clarity (Figure 1A).

The major issue raised in the generation of our ZIKV Replicon is the presence of the CMV promoter upstream of the 5' UTR in the amplicon Z1-PURO. We are aware that by the ISA method, ZIKV genomic RNA molecules are produced after recombination of DNA fragments whose transcription depends on the use of the CMV promoter. However, after a first round of transcription/translation from recombinant DNA, an autonomous replication system is set up from the double-stranded RNA. We assume that a classical transcription-translation process, dependent on CMV (due to the residual transfection entry) exists in the early stages. We therefore provide the following new points to remove any ambiguity surrounding the fact that effective autonomous replication of the ZIKV genomic RNA molecules does occur and that CMV promoter is neutral in the system, with regard to the regulations identified on HO-1.

- First, after puromycin selection of cells expressing and replicating the replicon, (puro resistant and GFP positive) the presence of viral genomic RNA was verified by RT PCR. Please note that amplification signals have been obtained for parts of the replicon encoding either the GFP or the viral genes (NS3 and NS5) (Figure 1A and 1B). We would like to point out that the PCR amplification signals presented in Figure 1B are representative of the results obtained over the duration of the experiment (from passages 1 to 5). We did not note any significant variation in the quantity of these signals. This suggests that a stabilized replicon system, sufficient to maintain puromycin resistance, is being expressed and does not lead to an increase in viral genomic RNA over time.

- Second, on your recommendation, replicon expression was verified by fluorescence monitoring of "GFP positive cells" (Figure 1C and Figure 1D added) as well as by immunodetection of double-stranded replicative RNAs with a J2 antibody (Figure 1D, added). An immunodetection of non-structural viral proteins (NS1) was also performed and confirmed the expression of the replicon at the protein level (Supplemented data Fig S1).

We have provided additional indirect evidence of the maintenance of a replicative viral genomic RNA in HEK-293A cells. These modifications are added to section 3.2 of the revised manuscript and described below:

- the effect of ribavirin, a nucleoside analog that blocks viral RNA synthesis significantly reduces the % of GFP cells (Figure 1C in the revised version). The use of this inhibitor (NITD008) has already been validated for the ZIKV replicon in previous studies (Xie et al, 2016, eBioMedecine and Li et al, 2018, two references added in the new version). A decrease of the GFP signal is therefore correlated with inhibition of the replicon replication.

- the ISRE/SEAP response that could be measured throughout the experiment is presented in Figure 1E.

However, the presence of CMV promoter and constitutive expression of mRNA complicate the use of replicon and interpretation of data reported in this study.

To address the question of the interpretation of the data, given the possible existence of constitutive expression of an mRNA via the CMV promoter, we conducted an experiment to examine if CoPP addition or the overexpression of HO-1 could affect the expression level of a CMV promoter driven gene. We then transfected HEK-293 with a plasmid containing GFP under the control of the pCMV using the plasmid pIRES2-GFP. Under both conditions (CoPP and overexpressed HO-1) we did not observe any decrease in GFP expression and the same observation was made with ribavirin (Cytometry results and Western blot analysis of HO-1 in Supplemental Figure S2A and 2B). HO-1 does not have the ability to decrease the constitutive expression of a gene under CMV promoter and/or IRES dependent translation. The results observed in Figure 2 are therefore most likely related to an effect of HO-1 on ZIKV replicon.

It will be useful if the authors used WT virus in the experiment discussed in figure 2. A more reliable conclusion can be made with respect to the overexpression of HO-1 in cell lines followed by virus infection. Since there is conflicting data from replicon and virus, in order to have reliable results, use virus and replicon in the same assay system and compare results. As is, the authors fail to confirm that HO-1 is down regulating ZIKV replication.

To answer this point and provide a more reliable conclusion, we now present an “infection experiment” in the same assay system i.e. the HEK-293 and with the use of a recombinant virus ZIKV-GFP (supplemental Figure S2C and S2D). Overexpression of HO-1 or its induction with CoPP lead to a decrease in the number of infected cells (GFP positive cells followed in cytometry, Figure S2C). We can observe with the Western Blotting results presented in Figure S2-D that HO-1 is not detectable in the control. Following infection with ZIKV, an increased level in HO-1, with CoPP and/or overexpression due to the presence of the plasmid pcDNA3.1-HO-1-FLAG can be correlated with a reduced signal of the ZIKV E protein.

Fig 4:  The effect of ZIKV MOI in controlling the transcription of HO-1 is not clear. There is RNA expression at MOI 1-5. For quantitative analysis, authors will have to use qRT to determine the RNA copy numbers of viral RNA and HO-1 RNA. As is, it does not look like transcriptional control. Similarly, in Fig 3B there is mRNA expression of HO-1 but no protein is seen in 3A. Along with the data shown in S2, the protein expression of HO-1 is negatively affected

We have modified these figures in accordance with the recommendations stated and added quantitative analysis as suggested (q-PCR ddCT values): Figure 3C, Figure 3F and Figure 4C.

We agree that the effects of ZIKV replication on HO-1 can take place at the transcriptional level but also at post-transcriptional or post-translational level as highlighted by the increase effect on HO-1 protein compared to HO-1 mRNA. We have modified the discussion to better explain these differences.

We thank you again for the feedback. We believe that our additional work and modifications of the manuscript have greatly improved the presented work, providing a much more thorough validation of the experimental systems that we used.

Reviewer 2 Report

Kalamouni et al present experiments showing that HO-1 and its induction by coPP inhibits ZIKV. Moreover, the authors show evidence that ZIKV inhibits the expression of HO-1. Overall the study appears to be well done, but there are some points the authors should address to improve the manuscript.

1.      Comparison of level of CoPP-mediated HO-1 induction in figure 1A with that shown in figure 4A in the absence of infection suggests that HO-1 induction is stronger in cells harboring the ZIKV replicon. Have the authors performed the same CoPP titration experiment shown in figure 1A in naïve cells without the ZIKV replicon?

2.      The Western blot and RT-PCR analyses shown in figures 3 and 4 show a single replicate for each condition. The authors should indicate how many times these experiments were performed and provide quantitative and statistical analyses where appropriate. Ideally the RT-PCR would be performed using a more quantitative approach (real-time PCR). Do the authors have access to this approach?

Minor comments:

1.      The authors conclude that HO-1 is affected by ZIKV at the “transcriptional” level based on RT-PCR analysis. This method measures transcript levels which is affected by by mRNA synthesis and decay. The authors should modify their interpretation accordingly.

2.      In panel 2C the overexpression is shown with anti-flag antibody. It would be more informative to probe with ant-HO-1 antibody as this would indicate the level of overexpression achieved.

3.      In panel 2B the authors should report the concentration of ribavirin in molarity.

4.      On line 270 the authors state that HO-1 modulates ZIKV replication. Technically the effect could be through replication (RNA synthesis) or translation (protein synthesis). The authors revise to note this.

5.      On lines 46-47 the authors state that the virus genome is translated into a single polyprotein which is then cleaved into mature proteins. This is not accurate as processing of most viral proteins occurs co-translationally.

6.      My opinion is that the virus titer data shown in the supplementary data should be included in the main figures.

Author Response

Reviewer 2:

Thank you for the feedback and the opportunity to improve the understanding and presentation of our data. We propose a revised version of our manuscript with additional experimental data that we believe addresses the major concerns raised.

Kalamouni et al present experiments showing that HO-1 and its induction by coPP inhibits ZIKV. Moreover, the authors show evidence that ZIKV inhibits the expression of HO-1. Overall the study appears to be well done, but there are some points the authors should address to improve the manuscript.

1. Comparison of level of CoPP-mediated HO-1 induction in figure 2A with that shown in figure 4A in the absence of infection suggests that HO-1 induction is stronger in cells harboring the ZIKV replicon. Have the authors performed the same CoPP titration experiment shown in figure 1A in naïve cells without the ZIKV replicon?

We would first like to thank the reviewer for their comments.

For this first concern, we remind that induction of HO-1 by CoPP is observed in two different cell lines: HEK 293 in Figure 2 and A549 in Figure 4A. We noticed that we have on one hand HEK293A cells that do not constitutively express a high level of HO-1 and on an other hand, A549 cells that have a higher basal level of HO-1compared to HEK293. To illustrate our point, we tested the expression of HO-1 in the three lines used in our study, HEK 293, A549 and VERO. Please see below. We would like to point out that HEK293 presented below are ‘naïve’ cells without replicon. Like for the HEK293 with the ZIKV Replicon, the basal level of HO-1 is not detectable.

2.      The Western blot and RT-PCR analyses shown in figures 3 and 4 show a single replicate for each condition. The authors should indicate how many times these experiments were performed and provide quantitative and statistical analyses where appropriate. Ideally the RT-PCR would be performed using a more quantitative approach (real-time PCR). Do the authors have access to this approach?

We are very sorry that we did not specify this point in the first version of the paper. All of these experiments were repeated at least three times and we chose a representative panel of these repeats in Figures 3 and 4. In addition, we have taken your comment into account and have presented the statistical analyses of q-PCR data in the new Figures 3 and 4.

 Minor comments:

1.      The authors conclude that HO-1 is affected by ZIKV at the “transcriptional” level based on RT-PCR analysis. This method measures transcript levels which is affected by mRNA synthesis and decay. The authors should modify their interpretation accordingly.

This point has been addressed in the new version. New results presented in the updated figure 3 show the qPCR values obtained for HO-1. We do agree that the reduction observed may be the result of a change at the “transcriptional level” but also in the decay of the messenger.

2. In panel 2C the overexpression is shown with anti-flag antibody. It would be more informative to probe with ant-HO-1 antibody as this would indicate the level of overexpression achieved.

On your recommendation, we performed an immunodetection with an anti HO-1 and modified the figure accordingly.

3. In panel 2B the authors should report the concentration of ribavirin in molarity.

This has been changed in the new version.

4. On line 270 the authors state that HO-1 modulates ZIKV replication. Technically the effect could be through replication (RNA synthesis) or translation (protein synthesis). The authors revise to note this.

We fundamentally agree that the effects of HO-1 on the viral replication can take place both at the level of RNA synthesis and/or protein synthesis.

5. On lines 46-47 the authors state that the virus genome is translated into a single polyprotein which is then cleaved into mature proteins. This is not accurate as processing of most viral proteins occurs co-translationally.

We have clarified this point in the text (line 52).

6. My opinion is that the virus titer data shown in the supplementary data should be included in the main figures.

As this experiment was performed with a different cell line (Vero), we preferred not to integrate the data into the main part of the manuscript. These remarks are obviously relevant and we can explain some choices. We conducted the replicon and infection experiments separately with two different cell lines for the following reason: We used HEK-293A for implementation of a ZIKV replicon but found that they were poorly permissive to the virus. We therefore used cells known to be easily infected with ZIKV (the A549) (around 30% with 107 PFU/mL at 24h PI) to study the effect of infection on HO-1 induction. But we have ensured that despite a percentage of infection in the order of 15% with 103 PFU/mL at 24h PI, in the 293 cells, we also observe a decrease in the induction of HO-1 by CoPP in this model (Figure S2 C).

Round 2

Reviewer 1 Report

The revised manuscript with changes is acceptable for publication.